# Low-rank Prompt Interaction for Continual Vision-Language Retrieval

## ABSTRACT

Research on continual learning in multi-modal tasks has been receiving increasing attention. However, most existing work overlooks the explicit cross-modal and cross-task interactions. In this paper, we innovatively propose the **L**ow-rank **P**rompt **I**nteraction (LPI) to address this general problem of multi-modal understanding, which considers both cross-modal and cross-task interactions. Specifically, as for the former, we employ multi-modal correlation modules for corresponding Transformer layers. Considering that the training parameters scale to the number of layers and tasks, we propose Low-rank Interaction-augmented Decomposition to avoid memory explosion while enhancing the cross-modal association through sharing and separating common-specific low-rank factors. In addition, due to the multi-modal semantic differences carried by the low-rank initialization, we adopt hierarchical low-rank contrastive learning to ensure training robustness. As for the latter, we initially employ a visual analysis and identify that different tasks have clear distinctions in proximity. Therefore, we introduce explicit task contrastive constraints in the prompt learning process based on task semantic distance. Experiments on two retrieval tasks show performance improvements with the introduction of a minimal number of parameters, demonstrating the effectiveness of our method.

## CCS CONCEPTS

• **Information systems** → **Multimedia and multimodal retrieval**.

## KEYWORDS

Multi-modal, vision-language retrieval, continual learning, prompt learning

## 1 INTRODUCTION

The vision-language retrieval task requires us to train models to solve various retrieval challenges, including text retrieval, image retrieval, video retrieval, and referring expression comprehension. Addressing multi-modal downstream tasks under a continual learning setting aligns more closely with real-world demands. The model needs to continuously learn on tasks with non-stationary data distributions, which can lead to overfitting on current tasks and consequently cause catastrophic forgetting [23].

*ACM MM, 2024, Melbourne, Australia*

© 2024 Copyright held by the owner/author(s). Publication rights licensed to ACM.

ACM ISBN 978-x-xxxx-xxxx-x/YY/MM

https://doi.org/10.1145/nnnnnnn.nnnnnnn

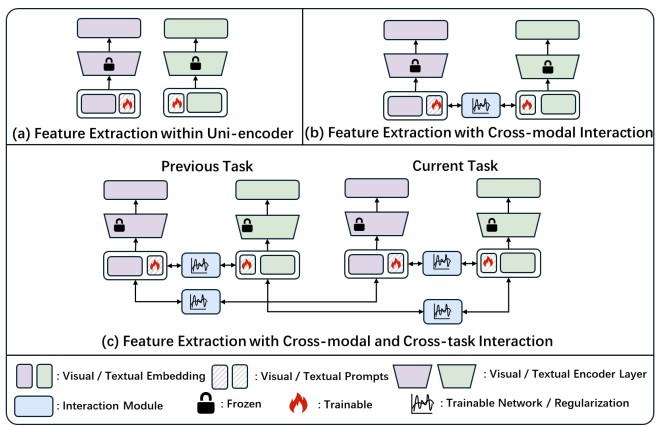

**Figure 1: Three Paradigms of Prompt Tuning in the Continual Learning Scenario.**

Traditional solutions to address catastrophic forgetting can be categorized into three types. The replay-based approach involves storing samples from previous tasks in a memory bank for training on future tasks. However, this can lead to data leakage and security issues, making it unacceptable in certain scenarios. The regularization-based approach seeks to prevent model forgetting by regularizing parameter updates, but it can only mitigate forgetting to a certain extent. The architecture-based approach allows for the independent learning of parameters for each task, which introduces many additional parameters.

Recently, prompt-based learning has made significant progress in the field of natural language processing (NLP). Inspired by this technology, some works have incorporated prompt techniques within pre-trained multi-modal models. Pre-trained vision-language models, such as CLIP (Contrastive Language-Image Pre-Training) [24] and GLIP (Grounded Language-Image Pre-training) [16] are trained on web-scale data, exhibit strong generalization capabilities. When we need to employ the model for downstream tasks, full model fine-tuning consumes a significant amount of computational and storage resources. By freezing the pre-trained model and training only a small number of prompt tokens, the model can achieve satisfactory results. We summarize the application of prompt learning in continual learning into three paradigms, as illustrated in Fig. 1. Paradigm (a) uses the uni-encoder to extract features, Paradigm (b) considers the cross-modal interaction, and Paradigm (c) considers both cross-modal and cross-task interactions.

Works [35, 36] have been developed for continual learning scenarios, however, numerous aspects could be improved. Such approaches are confined to single-modality tasks, such as image classification, and they have not taken into account the interactions between different modalities and tasks, thus failing to fully leverage

the performance of pre-trained models. Works [12, 19] have incorporated the cross-modal interaction within single-task scenarios, demonstrating the effectiveness of the interaction between different modalities. However, the introduction of interaction modules results in a substantial increase in the number of parameters, which becomes unacceptable as the number of tasks grows. How can we reduce the introduction of parameters while still incorporating interaction modules? In large language models, LoRA (Low-Rank Adaptation) [9] compress updates to network parameters. Inspired by this approach, we can employ a similar strategy to compress all learnable parameters within our model.

Taking into consideration the issues discussed above, we adhere to the third paradigm shown in Fig. 1 (c) to propose a novel **Low-rank Prompt Interaction (LPI)** method which accounts for both the cross-modal and cross-task interactions. For the cross-modal interaction, we integrate a low-rank interaction decomposition module, utilizing a shared low-rank factor to construct high-dimensional visual and textual prompts, which not only establishes cross-modal prompt associations but also reduces the number of introduced parameters. Since the low-rank interaction decomposition does not account for the hierarchical relationships among prompts, we introduce hierarchical prompt contrastive learning to achieve the hierarchical alignment of cross-modal prompts. Furthermore, we employ a cross-modal prompt fusion module to achieve the integration of cross-modal information within visual and textual prompts from different layers. For the cross-task interaction, we compute semantic distances across various tasks, classifying tasks into positive or negative samples based on the closeness of these distances in the semantic space. By applying contrastive learning, positive samples are drawn closer while negative samples are pushed further apart.

We conducted experiments on two retrieval tasks: image-text retrieval and referring expression comprehension. The results demonstrate that our method outperforms other approaches in the class-incremental learning setting [30], thus proving the efficacy of our method. In summary, our contributions are as follows:

- We propose Low-rank Prompt Interaction, a novel method that considers the cross-modal and cross-task interactions. The cross-modal interaction facilitates interaction between modalities, achieves hierarchical prompt alignment, and reduces introduced parameters while the cross-task interaction constrains prompt updates based on task semantic distance.
- In the class incremental setting, we conduct experiments on two complex vision-and-language tasks: image-text retrieval and referring expression comprehension. Our experimental results surpass the state-of-the-art approaches, demonstrating the efficacy of our method.

## 2 RELATED WORK

### 2.1 Continual Learning

Within the context of continual learning, an intelligent system is expected to acquire a sequence of knowledge. Unlike traditional models that are designed to learn from a single data distribution, these models are required to adapt to dynamically changing data distributions. A significant challenge arises if the model overfits on a particular task, leading to catastrophic forgetting, a significant decline in performance on previously learned tasks. To address

this problem, common methods include: (1) rehearsal-based methods [5, 26–29] that preserve representative or pseudo-samples to avoid catastrophic forgetting; (2) regularization-based methods [1, 13, 40]that constrain parameter updates; (3) architecture-based methods [14, 37] that allocate distinct parameters for each task to learn independently. Recent works [34–36] employ prompt-based learning techniques to train unique prompts for each task. These approaches have only been applied to simple uni-modal tasks and do not take into account the interactions between modalities or tasks. Consequently, they have not fully leveraged the potential of the pre-trained model's performance in more complex, multi-modal application scenarios.

### 2.2 Pre-trained Vision-Language Models

The Transformer architecture model has achieved tremendous success in natural language processing [3, 25, 33]. Subsequently, works [4, 8] have demonstrated their effectiveness in the field of computer vision as well, effectively establishing connections between multiple modalities. Due to its powerful representational capabilities and flexible structure, the Transformer-based architecture has become the ideal choice for linking multi-modal data. Following this, numerous effective pre-trained vision-language models like CLIP [24], GLIP [16], ALIGN [10] and Florence [39] have emerged. These models are trained on extensive datasets, learning rich joint visual-language representations, and exhibit commendable performance on zero-shot and few-shot learning. By employing fine-tuning techniques, these pre-trained models can be applied to specific downstream tasks, such as image-text retrieval, visual question answering, and more, finding widespread application across both the academic and industrial sectors.

### 2.3 Prompt Learning

Prompt learning was initially applied in the field of natural language processing [15, 17], enhancing the performance of pre-trained models through artificially designed or learnable prompts. Works [2, 11] introduces prompt learning into computer vision by adding prompts to visual input. Works like CoOp [42] and CoCoOp [41] apply it to Vision-Language (V-L) models. Furthermore, works like L2P [36], DualPrompt [35] and S-Prompts [34] adopt prompt learning in the continual learning setting. To further boost model performance, MaPle [12] and DCP [19] integrate prompts into both the visual and language encoders, taking into account the cross-modal interaction of prompts. In this work, we make full use of prompt tuning to maximize the performance of pre-trained vision-language models in the continual learning setting.

## 3 PRELIMINARY

### 3.1 Problem Formulation

We concentrate on tackling the problem of multi-modal continual learning for vision-language retrieval in the class-incremental learning(Class-IL) scenario [30]. Specifically, we conduct experiments on text-image retrieval and referring expression comprehension tasks. For the image-text retrieval task, we need to retrieve images based on captions while retrieving captions based on images. For the referring expression comprehension task, given a

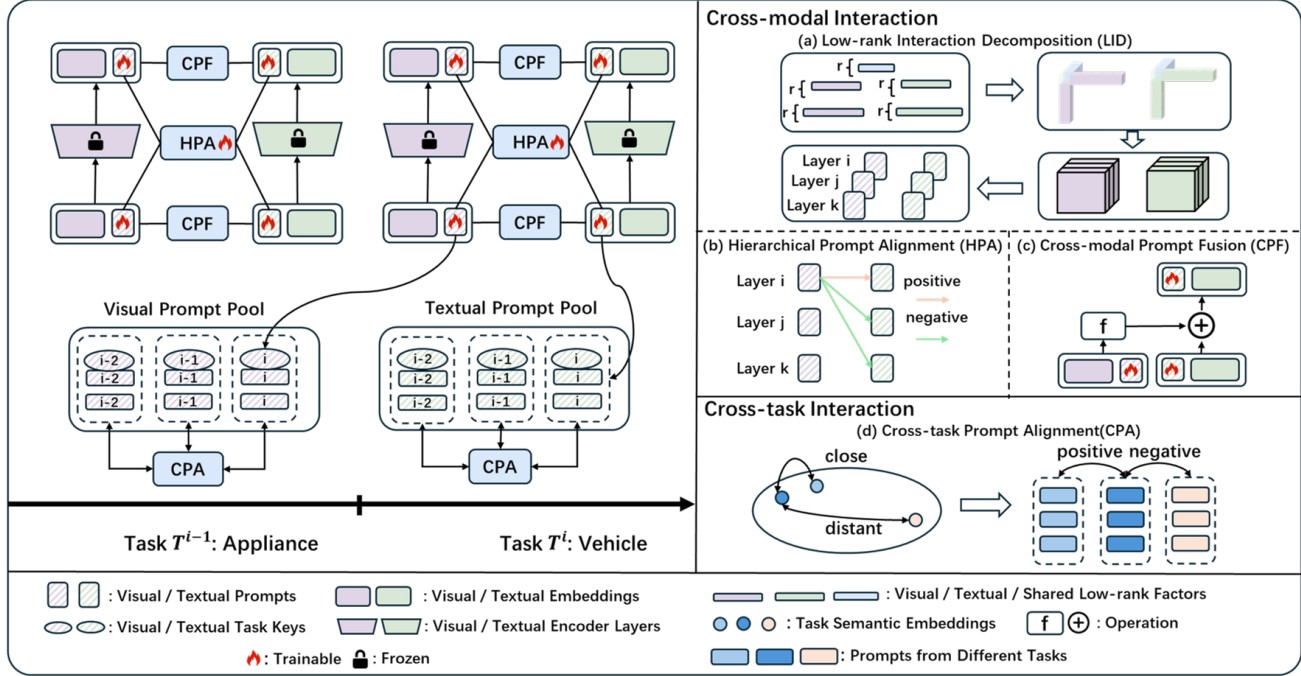

Figure 2: Illustration of the proposed LPI method. LPI consists of two parts, the cross-modal and cross-task interactions. As for the cross-modal interaction, we introduce three modules, low-rank interaction decomposition (LID), hierarchical prompt alignment (HPA), and cross-modal prompt fusion (CPF). LID decomposes high-dimensional prompts into low-rank vectors. HPA utilizes contrastive learning to align prompts across different layers. CPF achieves cross-layer information fusion between modalities. As for the cross-modal interaction, we introduce the cross-task prompt alignment (CPA) module which aligns prompts with task semantic distance. For each task, our method learns unique visual and textual prompts and interact modules. We store the prompts in the visual and textual prompt pool with task-specific keys which are used for predicting task identity during the inference period. As the task identity is unknown during the inference stage, we employ a method similar to that used in S-Prompts [34] to predict the task identity.

caption and an image, we need to locate the position of the content described by the caption within the image.

Our model is designed to progressively acquire knowledge across a sequence of tasks, with the ultimate goal of achieving comprehensive expertise in all tasks. Formally, the model sequentially acquires knowledge on a series of tasks, denoted as $T = \{T^1, T^2, ..., T^K\}$, where $K$ represents the overall count of tasks seen. For each task $T^k$, we denote $\mathbb{D}_k = \{v_i^k, q_i^k, a_i^k\}_{i=1}^{N_k}$ as the available data with $N_k$ samples, where $v_i^k$, $q_i^k$ is the $i$-th input vision and language representation respectively and $a_i^k$ is the corresponding label. During training period, given visual input $v_i$ and textual input $q_i$, we can compute the task-specific output $o_i = MODEL(v_i, q_i)$, $\mathcal{L}_{base} = \mathcal{L}_{problem}(o_i, a_i)$ where $\mathcal{L}_{problem}$ is the task-specific loss. For the image-text retrieval task, $\mathcal{L}_{problem}$ is the cross-entropy loss between logits and the truth label. For the referring expression comprehension, $\mathcal{L}_{problem}$ is the same as that in GLIP. During the testing phase, we evaluate the model's performance on both the current and all previously learned tasks.

## 3.2 Base Network

Our proposed method is based on the pre-trained vision-language models, CLIP for the image-text retrieval task and GLIP for the referring expression comprehension task. Both CLIP and GLIP can be extracted as a vision encoder, a language encoder, and a downstream head. We omit the downstream head here for clarity. We only make improvements at the feature extraction stage, while maintaining consistency in structure and parameters for the rest of the model. Both the vision encoder and the language encoder have $N_L$ transformer layers, denoted as $\{\mathcal{V}_i(\cdot)\}_{i=0}^{N_L-1}$ for vision encoder and $\{\mathcal{L}_i(\cdot)\}_{i=0}^{N_L-1}$ for language encoder. The feature embedding dimension of the vision encoder is $d_v$, while the feature embedding dimension of the language encoder is $d_l$. Visual input $v_i^k$ is divided into patches and projected into patch embeddings as the vision encoder input. Textual input $q_i^k$ is tokenized and projected into word embeddings as the language encoder input.

## 3.3 Deep Prompting

Deep prompting attaches learnable visual and textual prompts to the input of the first $D$ transformer layers.

**Deep Prompting.** Deep vision prompting and deep language prompting adopt the same strategy. To learn the modality prompts, we introduce learnable tokens $\{P_i^m \in \mathbb{R}^{L_m \times d_m}\}_{i=1}^D$ where m represents v(ision) or l(anguage) modality, $L_m$ is the prompt length and $d_m$ is the prompt dimension. The input token can be denoted as $[E_0^m, W_0^m]$. $E_0^m$ and $P_0^m$ have the same token length and embedding dimension. For the first D transformer layer, the forward process can be formulated as:

$$
\begin{aligned}
I_{i-1}^m &= P_{i-1}^m + E_{i-1}^m, \\
[E_i^m, W_i^m] &= \mathcal{M}_{i-1}([I_{i-1}^m, W_{i-1}^m]), \\
i &= 1, 2, \cdots, D.
\end{aligned} \tag{1}
$$

For the rest transformer layers, the forward process can be formulated as:

$$
[E_i^m, W_i^m] = \mathcal{M}_{i-1}([E_{i-1}^m, W_{i-1}^m]), \quad i = D+1, \cdots, N_L, \tag{2}
$$

where $\mathcal{M}$ represents $\mathcal{V}$ (vision encoder) or $\mathcal{L}$ (language encoder).

# 4 LPI: LOW-RANK PROMPT INTERACTION

Our proposed method, Low-rank Prompt Interaction is illustrated in Fig. 2. Concretely, LPI consists of two parts, the cross-modal and cross-task interaction. As for the cross-modal interaction, we introduce three modules, low-rank interaction decomposition in Sec. 4.1, hierarchical prompt alignment in Sec. 4.2, and cross-modal prompt fusion Sec. 4.3. As for the cross-modal interaction, we introduce the cross-task prompt alignment in Sec. 4.4. Finally, we introduce the training details in Sec. 4.5.

## 4.1 Low-rank Interaction Decomposition

The Low-rank Interaction Decomposition decomposes the high-dimensional prompts into multiple low-rank vectors, which is shown in Fig. 2 (a). To establish the interaction between modalities, we have the visual prompts and textual prompts share the same low-rank vector, which not only facilitates the cross-modal interaction but also compresses the number of parameters introduced.

For a three-dimensional vector $\mathbf{P} \in \mathbb{R}^{i \times j \times k}$, we can decompose it into three low-rank two-dimensional vectors, $D_1 \in \mathbb{R}^{i \times r}$, $D_2 \in \mathbb{R}^{j \times r}$ and $D_3 \in \mathbb{R}^{k \times r}$. $\mathbf{P}$ can be formulated as:

$$
\mathbf{P}[i][j][k] = AVG(D_1[i]@D_2[j]@D_3[k]), \tag{3}
$$

where @ means the dot product and [i] means the index $i$ of the specific dimension.

The visual prompts $PV \in \mathbb{R}^{D \times L^v \times d^v}$ in the vision encoder are decomposed into $D_1, D_2^v$ and $D_3^v$ and textual prompts $PL \in \mathbb{R}^{D \times L^l \times d^l}$ in the language encoder are decomposed into three low-rank factors $D_1, D_2^l$ and $D_3^l$. The visual prompts and textual prompts share the same low-rank factor $D_1$, thus establishing the interaction between visual and textual modality.

## 4.2 Hierarchical Prompt Alignment

As low-rank interaction decomposition fails to consider the hierarchical relationships among prompts from different transformer layers, we design Hierarchical Prompt Alignment (HPA) to mitigate the multi-modal semantic discrepancy between different layers.

As shown in Fig. 2 (b), the objective of HPA is to perform contrastive learning over the learnable visual and textual prompts

between different transformer layers. We consider visual prompts and text prompts that are on the same layer as positive samples, and those that are not on the same layer as negative samples, utilizing the cross-entropy loss function to calculate the semantic disparity of cross-modal prompts. For $PV = \{P_1^v, P_2^v, \cdots, P_D^v\} \in \mathbb{R}^{D \times L_v \times d_v}$ and $PL = P_1^l, P_2^l, \cdots, P_D^l \in \mathbb{R}^{D \times L_l \times d_l}$, we first calculate the average values within the last dimension,

$$
\tilde{PV} = AVG(PV), \quad \tilde{PL} = AVG(PL). \tag{4}
$$

We set $L_v = L_l$. At this point $\tilde{PV}$ and $\tilde{PL}$ have the same shape, thus we can compute their score matrix $MS$ and cross-modal loss $\mathcal{L}_{modal}$,

$$
MS = (\tilde{PV}/\tau_{modal})(\tilde{PL}^T/\tau_{modal}),
$$

$$
\mathcal{L}_{modal} = -\frac{1}{D}\sum_{i=1}^{D}\sum_{j=1}^{D} y_{ij} \log MS_{ij}, \quad y_{ij} = \begin{cases} 0 & i \neq j, \\ 1 & i = j. \end{cases} \tag{5}
$$

where $\tau_{modal}$ represents the temperature for the hierarchical prompt alignment module and $y$ is the corresponding label.

## 4.3 Cross-modal Prompt Fusion

The LPI module establishes cross-modal associations, while the HPA accomplishes hierarchical alignment. Furthermore, we introduce the Cross-modal Prompt Fusion (CPF) to achieve cross-layer information fusion, which involves considering the outputs of previous layers from different modalities during the forward propagation process of the encoder. We conduct the cross-modal prompt fusion across the remaining $D-1$ layers except for the first layer. The specific interaction process is shown in Fig. 3.

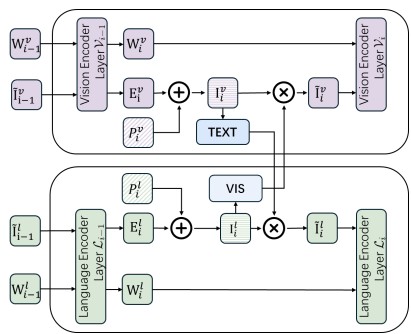

**Figure 3: Interaction Module.**

Before feeding the $I_i^v$ and $I_i^l$ into the encoders, we facilitate the cross-modal interaction and employ a fully connected layer to perform cross-modal mapping, thereby projecting different modalities onto the same dimensional space. To take previous prompt information into account, we combine the interactive information with the output from the preceding layer using momentum summation,

$$
\begin{aligned}
\tilde{I_i^v} &= \lambda_v I_i^v + (1-\lambda_v)VIS(I_i^l), \\
\tilde{I_i^l} &= \lambda_l I_i^l + (1-\lambda_l)TEXT(I_i^v), \\
i &= 1, \cdots, D-1,
\end{aligned} \tag{6}
$$

where $VIS$ is the language to vision projection layer, $TEXT$ is the vision to language projection layer, $\lambda_v$ is the visual momentum and

$\lambda_l$ is the textual momentum. Then $I_i$ will be replaced with $\tilde{I}_i$ as input to the transformer layer.

To avoid the introduction of an excessive number of parameters by the linear layer, we use the same decomposition strategy in Sec. 4.1 to decompose $p_t$ into three low-rank factors. As for the parameters in the interaction module, assuming the input dimension is $d_{in}$ and the output dimension is $d_{out}$. The weight can be denoted as $p_w \in \mathbb{R}^{(D-1) \times d_{in} \times d_{out}}$ and the bias can be denoted as $p_b \in \mathbb{R}^{(D-1) \times d_{out}}$. Thus the total learnable parameters can be denoted as $p_t \in \mathbb{R}^{(D-1) \times (d_{in}+1) \times d_{out}}$. Thus we can decompose $p_t$ into $D_1 \in \mathbb{R}^{(D-1) \times r}$, $D_2 \in \mathbb{R}^{(d_{in}+1) \times r}$ and $D_3 \in \mathbb{R}^{d_{out} \times r}$.

## 4.4 Cross-task Prompt Alignment

When training on task $T^k$, we utilize the word embedding to convert the task name seen, denoted as $N_1, N_2, \cdot, N_k$, to vectors as the task semantic embeddings. Using the visualization technique, we observe a clear distance relationship between tasks, which is shown in Fig. 4.

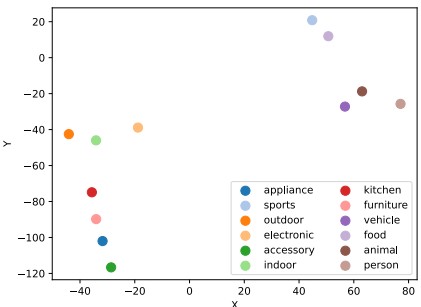

**Figure 4: Task Distance Visualization using t-SNE [32].**

Subsequently, we calculate the cosine similarity for each task's semantic embeddings. Pairs of tasks with cosine similarity greater than and equal to a predefined threshold are considered positive tasks, while those below the threshold are treated as negative tasks, we calculate the label $z$ for cross-entropy loss as follows:

$$S_i = EMBEDDING(N_i),$$
$$c_{ij} = COS(S_i, S_j),$$
$$z_{ij} = \begin{cases} 0 & c_{ij} < threshold, \\ 1 & c_{ij} > threshold. \end{cases} \quad (7)$$

We denote $PV_i$, $PL_i$ as the prompts for task $T^i$. Prompts in positive tasks are considered positive of each other, and prompts in negative tasks are considered negative of each other. Thereby, we can calculate visual score $TS^v$ and textual score $TS^l$ among different tasks, For visual prompts and textual prompts in task $T_k$, We first merge the second and third dimensions, stretching each prompt into a one-dimensional vector. We then compute the scores between visual prompts and textual prompts across different tasks,

$$TS_{ij}^v = SUM(PV_i PV_j^T),$$
$$TS_{ij}^l = SUM(PL_i PL_j^L). \quad (8)$$

Using the visual and textual scores, we can calculate the visual and textual losses $L^m$. Considering the numerical imbalance between positive and negative samples, we employ the NT-BXent (Normalized Temperature-scaled Binary cross-entropy) [7] loss function for computation, which is based on NT-Xent (Normalized Temperature-Scaled Cross-Entropy) [6]. Cross-task prompt loss between task $T^i$ and $T^j$ can be presented as follows:

$$l_{ij}^m = -z_{ij} \log \sigma(TS_{ij}^m / \tau_{task}) - (1 - z_{ij}) \log \sigma((1 - TS_{ij}^m) / \tau_{task}), \quad (9)$$

where $\tau_{task}$ represents the temperature for the cross-task interaction, $\sigma$ stands for the Sigmoid function, Thus we can compute the loss $L_i^m$ for each task $T^i$,

$$L_i^m = \frac{1}{N_{pos}} \sum_{j=1}^{k} z_{ij} l_{ij}^m + \frac{1}{N_{neg}} \sum_{j=1}^{k} (1 - z_{ij}) l_{ij}^m, \quad (10)$$

where $N_{pos}$ and $N_{neg}$ are positive and negative sample number of task $T^i$ respectively. The final cross-task loss $\mathcal{L}_{task}$ can be formulated as following:

$$\mathcal{L}_{task} = \frac{\sum_{i=1}^{k} L_i^v + \sum_{i=1}^{k} L_i^l}{2k}. \quad (11)$$

## 4.5 Training

We combine the problem-specific loss, the cross-modal loss and cross-task loss to train the model, i.e.,

$$\mathcal{L}_{our} = \lambda_1 \mathcal{L}_{base} + \lambda_2 \mathcal{L}_{modal} + \lambda_3 \mathcal{L}_{task}, \quad (12)$$

where $\lambda_1, \lambda_2$ and $\lambda_3$ are set to 0.8,0.1,0.1 to balance the three losses. Additional training and inference details are provided in the supplementary material.

## 5 EXPERIMENTS

In this section, we perform experiments on two vision-language retrieval tasks under the same class-incremental learning scenario [31], where we need to solve each task seen so far with task identity unknown. We compare our proposed LPI with state-of-the-art methods and conduct ablation studies.

### 5.1 Experiment Setting

*5.1.1 Model.* We propose two models, **LPI-M(ini)** and **LPI-P(ro)**.

**LPI-M:** The LPI-M employs the Low-rank Interaction Decomposition module to compress visual and textual prompts and incorporates Hierarchical Prompt Alignment and Cross-task Prompt Alignment during the training phase.

**LPI-P:** The LPI-P incorporates Cross-modal Prompt Fusion on top of LPI-M.

LPI-M can extract uni-modal features separately, and saving the uni-modal features of the retrieved objects in advance can enhance retrieval speed. LPI-P, on the other hand, carries out the cross-modal interaction, demonstrating higher performance.

*5.1.2 Dataset and Task Division.* We conduct experiments on two visual-language retrieval tasks, namely image-text retrieval and referring expression comprehension. For the image-text retrieval task, we select the **MS-COCO** [18] dataset, while for the referring expression comprehension task, we choose the **RefCOCO**[38], **RefCOCO+**[38], and **RefCOCOg**[22] datasets. We divide the whole

**Table 1: Performance Evaluation on Image-Text Retrieval. Bold: best results, Underline: second best results.**

| Method | Image-to-Text Retrieval | | | | Text-to-Image Retrieval | | | |
|---|---|---|---|---|---|---|---|---|
| | R@1($\uparrow$) | R@5($\uparrow$) | R@10($\uparrow$) | Forgetting($\downarrow$) | R@1($\uparrow$) | R@5($\uparrow$) | R@10($\uparrow$) | Forgetting($\downarrow$) |
| CLIP [24] | 67.87 | 87.71 | 93.78 | **0.71** | 49.38 | 78.55 | 88.45 | **2.17** |
| L2P [36] | 64.45 | 84.09 | 91.30 | 7.17 | 52.98 | 80.54 | 89.62 | 3.75 |
| S-Prompt [34] | 67.27 | 87.88 | 93.56 | 4.48 | 54.13 | 83.02 | 91.24 | 3.41 |
| LPI-M | **70.29** | **88.92** | **95.46** | 3.71 | **56.09** | **84.21** | **92.03** | 4.73 |

**Table 2: Performance Evaluation on Referring Expression Comprehension. Bold: best results, Underline: second best results.**

| Dataset | | Metric | GLIP-T (A) [16] | L2P [36] | S-Prompts [34] | MaPLe [12] | DCP [19] | LPI-M | LPI-P |
|---|---|---|---|---|---|---|---|---|---|
| RefCOCO | val | R@1($\uparrow$) | 29.58 | 29.65 | 30.48 | 27.93 | 27.45 | 31.27 | **33.00** |
| | | R@5($\uparrow$) | 74.10 | 74.54 | 77.72 | 79.12 | 79.28 | 82.05 | **83.55** |
| | | R@10($\uparrow$) | 81.87 | 81.62 | 88.96 | 90.44 | 89.95 | 90.72 | **91.39** |
| | | Forgetting($\downarrow$) | - | **0.38** | 1.85 | 3.54 | 2.86 | 2.45 | 2.38 |
| | testA | R@1($\uparrow$) | 29.62 | 29.65 | 33.50 | 28.33 | 35.34 | **37.38** | 37.17 |
| | | R@5($\uparrow$) | 76.00 | 74.92 | 74.24 | 76.95 | 71.98 | 82.33 | **84.56** |
| | | R@10($\uparrow$) | 81.54 | 80.73 | 82.62 | 82.05 | 80.20 | **92.01** | 91.95 |
| | | Forgetting($\downarrow$) | - | **-0.09** | 2.07 | 3.32 | 7.37 | 11.66 | 12.26 |
| | testB | R@1($\uparrow$) | 32.48 | 32.40 | 34.28 | 32.95 | 32.93 | 35.30 | **37.10** |
| | | R@5($\uparrow$) | 77.73 | 78.18 | 84.06 | 84.54 | 84.18 | 85.27 | **86.24** |
| | | R@10($\uparrow$) | 83.26 | 84.31 | 91.56 | 92.26 | 91.41 | 92.60 | **92.74** |
| | | Forgetting($\downarrow$) | - | **0.40** | 1.65 | 1.61 | 1.08 | 1.56 | 1.53 |
| RefCOCO+ | val | R@1($\uparrow$) | 29.77 | 29.67 | 29.73 | 27.78 | 27.97 | 30.99 | **31.72** |
| | | R@5($\uparrow$) | 72.43 | 72.61 | 79.07 | 79.78 | 80.47 | 80.43 | **82.76** |
| | | R@10($\uparrow$) | 79.05 | 79.03 | 89.28 | 90.06 | 89.75 | 90.86 | **91.81** |
| | | Forgetting($\downarrow$) | - | **0.46** | 2.46 | 3.28 | 3.28 | 2.37 | 2.53 |
| | testA | R@1($\uparrow$) | **34.32** | 31.31 | 33.89 | 31.96 | 32.11 | 33.03 | 30.20 |
| | | R@5($\uparrow$) | 73.91 | 74.29 | **75.94** | 75.72 | 72.05 | 75.20 | 74.95 |
| | | R@10($\uparrow$) | 81.88 | 81.95 | **86.63** | 81.83 | 80.59 | 84.77 | 84.31 |
| | | Forgetting($\downarrow$) | - | **0.44** | 6.59 | 4.03 | 15.31 | 4.69 | 3.29 |
| | testB | R@1($\uparrow$) | 32.15 | 31.95 | 35.91 | 33.49 | 32.82 | 35.98 | **37.02** |
| | | R@5($\uparrow$) | 75.63 | 76.67 | 83.75 | 85.23 | 83.87 | 84.68 | **86.52** |
| | | R@10($\uparrow$) | 82.07 | 83.20 | 92.00 | 92.09 | 91.05 | **93.52** | 93.32 |
| | | Forgetting($\downarrow$) | - | **0.24** | 2.22 | 2.14 | 2.07 | 2.04 | 1.40 |
| RefCOCOg | val | R@1($\uparrow$) | 41.81 | 42.07 | 39.32 | 36.79 | 35.47 | 40.94 | **42.30** |
| | | R@5($\uparrow$) | 81.49 | 81.45 | 83.84 | 79.88 | 78.86 | 84.48 | **84.66** |
| | | R@10($\uparrow$) | 85.94 | 85.73 | 89.85 | 88.71 | 87.03 | 91.09 | **91.68** |
| | | Forgetting($\downarrow$) | - | **0.34** | 3.20 | 4.89 | 4.83 | 4.07 | 4.26 |
| | test | R@1($\uparrow$) | **41.48** | 41.07 | 37.99 | 36.74 | 34.76 | 40.02 | 41.46 |
| | | R@5($\uparrow$) | 82.26 | 82.14 | 83.12 | 79.19 | 78.84 | **84.81** | 84.50 |
| | | R@10($\uparrow$) | 86.57 | 86.67 | 90.51 | 87.63 | 86.97 | 91.61 | **91.70** |
| | | Forgetting($\downarrow$) | - | **0.55** | 3.78 | 5.02 | 4.97 | 4.44 | 4.66 |

dataset into 12 categories according to the super category of the images, which are appliance, sports, outdoor, electronic, accessory, indoor, kitchen, furniture, vehicle, food, animal, and person. Further details can be found in the supplementary materials.

*5.1.3 Evaluation Metrics.* We conduct experiments under the class-increment setting, utilizing **Average Accuracy** and **Forgetting** as metrics following previous work [20, 21]. Specifically, we compute the Recall at K, denoted as $R@K$, which means the recall rate of samples containing the target within the top K results ranked by the predicted probability of retrieval. The forgetting rate, represented as $F@K$, is derived by subtracting the maximum value of $R@K$ from previous tasks from the value of $R@K$ obtained from the most recent task test. *Forgetting* is the average of $F@1$, $F@5$, and $F@10$.

*5.1.4 Implementation Detail.* We set the prompt length $L^v = L^l = 16$, $\tau_{modal} = 0.01$, $\tau_{task} = 0.01$, and $\lambda_v = \lambda_l = 0.9$. The prompt and interaction rank is set to 4. For the image-text retrieval task, all the training images are resized to $224 \times 224$. We select CLIP(ViT-B/16) as the backbone. We set $d^v = 768$, $d^l = 512$, $epoch = 5$, learning rate $lr = 0.01$. For the referring expression comprehension task, all the training images are resized to $448 \times 448$. We select GLIP-T(A) as the backbone. We set $d^v = 96$, $d^l = 768$, $epoch = 10$, learning rate $lr = 0.05$, $threshold = 0.4$. To train Our model, we use Adam optimizer with the cosine annealing scheduler. Further experimental details can be found in the supplementary material.

## 5.2 Performance Evaluation

*5.2.1 Baselines.* We compare our proposed method with pre-trained CLIP [24], GLIP [16], and prompt-based methods including L2P [36], S-Prompts [34], MaPLe [12], and DCP [19]. L2P and S-Prompts employ the uni-encoder for feature extraction. MaPLe and DCP account for the cross-modal interaction. Given that some of these methodologies are not originally tailored for a continual learning framework, we adapt them by incorporating their prompt tuning strategies.

*5.2.2 Image-text Retrieval.* The results of the image-text retrieval are shown in Tab. 1. Compared to CLIP, LPI-M shows improvements across all metrics, yet it has a higher *Forgetting*. In contrast to L2P, LPI-M demonstrates enhancements in every benchmark. LPI-M marginally surpasses S-Prompts in $R@1$, $R@5$, and $R@10$, while maintaining a similar *Forgetting*.

*5.2.3 Referring Expression Comprehension.* Tab. 2 presents the results of the referring expression comprehension. Our method shows significant improvements in $R@1$, $R@5$, and $R@10$, while its performance on the *Forgetting* is average. We observe that L2P has the lowest *Forgetting* in most tasks but limited improvement in *Recall*. In some tasks, its performance is even lower than the base model GLIP. This is attributed to training prompts for each task, where an incorrect prediction of task identity can mislead the inference. S-Prompts achieves a more balanced improvement in Recall compared to the base model. MaPLe and DCP show significant improvements in $R@5$ and $R@10$, though their $R@1$ is lower than GLIP on some test datasets.

DCP, LPI-M, and LPI-P exhibit a high *Forgetting* of up to 10% on testA of refcoco and refcoco+. A deeper analysis reveals that testA datasets have very few test samples in most tasks, leading to a significant drop in recall if even one sample is forgotten.

LPI-M and LPI-P surpass other comparative approaches on most metrics for the referring expression comprehension task. Specifically, LPI-M is well-suited for scenarios with a small number of training samples per task, while LPI-P is appropriate when there is an abundance of training samples for each task, as this allows for better learning of cross-modal and cross-task interactions, resulting in improved performance.

## 5.3 Ablation Study

We conduct ablation experiments on RefCOCO for the referring expression comprehension and calculate the metrics on the val dataset. We adopt the approach known as Decomposition Prompting (DP), which utilizes the low-rank interaction decomposition module. The method without prompt compression is referred to as Common Prompting (CP). Tab. 3 presents the metrics associated with the incorporation of different modules. TP indicates the type of prompting, DP or CP. The ablation experiment on prompt and interaction depth, as well as qualitative analysis, are provided in the supplementary material.

*5.3.1 Effectiveness of Different Modules.* We conduct an ablation study to explore the effectiveness of different modules, Hierarchical Prompt Alignment (HPA), Cross-modal Prompt Fusion (CPF), and Cross-task Prompt Alignment (CPA). The results are shown in the Tab. 3. It can be observed that HPA, CPF, and CPA all contribute to improvements in $R@1$, $R@5$, and $R@10$. Specifically, CPF allows the encoder to extract features more effectively by leveraging information from other modalities, which significantly enhances the $R@1$, $R@5$, and $R@10$. CPA reduces the distance between prompts of similar tasks, meaning that even if the task identity predicted during the inference stage is for a similar task, the prompts of these similar tasks still provide a certain level of guidance. Therefore, this increases the model's fault tolerance. The HPA achieves hierarchical alignment of encoders across different modalities, effectively constraining prompt updates and reducing the *Forgetting*.

We further conduct experiments on Low-rank Interaction Decomposition and find that not compressing the prompt results in very poor performance on $R@1$. Although there is a slight improvement in the $R@5$ and $R@10$ metrics, it leads to a higher *Forgetting*. Utilizing the CP strategy on LPI-P even underperforms the method that only uses DP. The analysis suggests that the prompts require more resources to train and random initialization cannot align well between modalities during the training process.

**Table 3: Ablation Study for Different Modules.**

| Module | | | | Metric | | | |
|---|---|---|---|---|---|---|---|
| HPA | CPF | CPA | TP | R@1 | R@5 | R@10 | Forgetting |
| - | - | - | - | 29.58 | 74.10 | 81.87 | - |
| - | - | - | CP | 27.10 | 76.04 | 87.25 | 3.19 |
| - | - | - | DP | 31.14 | 79.52 | 89.26 | 2.80 |
| - | - | ✓ | DP | 31.22 | 82.21 | 90.78 | 2.52 |
| - | ✓ | - | DP | 31.99 | 83.33 | 90.99 | 2.50 |
| ✓ | - | - | DP | 30.84 | 80.98 | 89.87 | 2.24 |
| ✓ | ✓ | - | DP | 31.74 | 83.76 | 91.30 | 2.65 |
| ✓ | - | ✓ | DP | 31.27 | 82.05 | 90.72 | 2.45 |
| - | ✓ | ✓ | DP | 32.34 | 83.37 | 91.05 | 2.52 |
| ✓ | ✓ | ✓ | CP | 27.72 | 77.68 | 88.90 | 3.13 |
| ✓ | ✓ | ✓ | DP | 33.00 | 83.55 | 91.39 | 2.38 |

*5.3.2 Rank.* We set the rank of the interaction module to 4 and vary the prompt rank at 1, 2, 4, 8, and 16. The results, as shown in Fig. 5 (a) and (b), indicate that $R@K$ initially increases and then decreases with the increase of the prompt rank, peaking at a rank of 4. Conversely, *Forgetting* follows the exact opposite pattern. We set the prompt rank to 4 and experiment with interaction rank at

1, 2, 4, 8, and 16, obtaining comparable results, as shown in Fig. 5 (c) and (d). Overall, as the interaction rank increases, $R@K$ initially rises and then falls, with a significant decrease in the $R@1$ only observed when the interaction rank is set to 8. As for the *Forgetting*, the value initially increases and then decreases, similar to that of prompt rank.

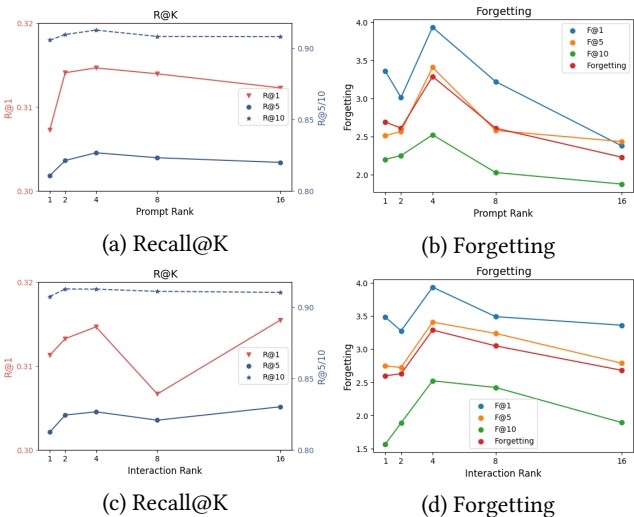

(a) Recall@K                   (b) Forgetting

(c) Recall@K                   (d) Forgetting

**Figure 5: Metrics with Different Prompt and Interaction Rank.**

## 5.4 In-depth Analysis

*5.4.1 Computational Complexity.* Tab. 4 presents the number of introduced parameters and computational requirements for different models. Our proposed model introduces a minimal number of parameters to facilitate the interaction between modalities compared with MaPLe and DCP. Notably, the computational requirement of LPI-M is consistent with that of L2P and S-Prompts.

**Table 4: Comparison of Computational Complexity.**

| Method | Param | Param% | Param%/Task | GFLOPS |
|---|---|---|---|---|
| L2P [36] | 5760 | 0.0038 | 0.0003 | 3349.41 |
| S-Prompts [34] | 0.17M | 0.1089 | 0.0091 | 3349.41 |
| MaPLe [12] | 9.30M | 5.7586 | 0.4799 | 3349.92 |
| DCP [19] | 88.36M | 36.7320 | 3.0610 | 3354.47 |
| LPI-M | 0.04M | 0.0285 | 0.0024 | 3349.41 |
| LPI-P | 0.148M | 0.0971 | 0.0081 | 3349.47 |

*5.4.2 Metrics for All Tasks.* Fig. 6 presents the metrics across 12 tasks for GLIP, Decomposition Prompting (DP) which solely employs the low-rank interaction decomposition module, LPI-M, and LPI-P. From the figure, it is evident that for the Recall metrics, LPI-P performs the best in most tasks, followed by LPI-M, then DP, with GLIP showing the lowest performance. In terms of the $R@1$ metric, the GLIP outperforms other methods in the appliance, person,

and furniture tasks. For the $R@5$ and $R@10$, the use of the prompt technique results in improvements across all tasks. Regarding the *Forgetting*, DP, LPI-M, and LPI-P exhibit the high *Forgetting* in sports, furniture, kitchen, accessory, and electronic tasks.

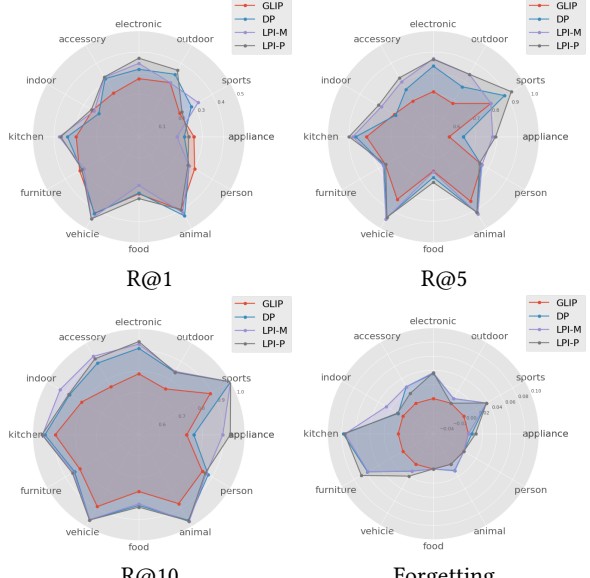

R@1                          R@5

R@10                         Forgetting

**Figure 6: Metrics for All Tasks.**

*5.4.3 Prompt Visualization.* We employ t-SNE [32] to visualize the visual and textual prompts learned by different methods. As is observed from the Fig. 7, prompts obtained through low-rank interaction decomposition are more clustered, whereas uncompressed prompts are quite dispersed.

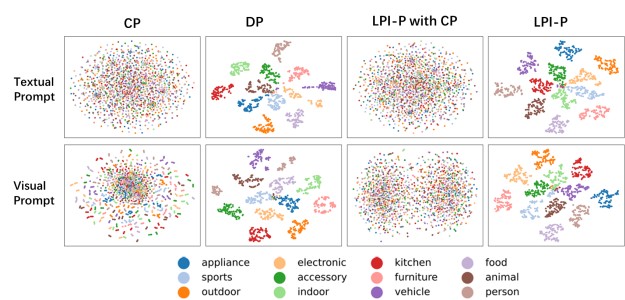

**Figure 7: Visual and Textual Prompt Visualization.**

## 6 CONCLUSION

In this paper, we propose a novel Low-rank Prompt Interaction method that considers both cross-modal and cross-task interactions. We further conduct experiments in the class-incremental setting on two vision-language tasks, image-text retrieval and referring expression comprehension. The results compared with state-of-the-art methods demonstrate the effectiveness of our method.

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
