# OpenReview forum: "Low-rank Prompt Interaction for Continual Vision-Language Retrieval"
_acmmm.org/ACMMM/2024/Conference — MM2024 Poster_

### Official Review · Reviewer_F7cT · 2024-05-24

**Rating:** 4
**Confidence:** 3

**Summary:**

This paper considers both cross-modal and cross-task interactions during continual learning and  proposes a novel method, LPI, which extends the prompt learning for multi-modal vision-langugae retrieval in the continual learning setting.
Overall, the paper is well written and easy to follow. Extensive experiments are conducted to show the effectiveness of the proposed method.

**Strengths:**

This paper considers both cross-modal and cross-task interactions during continual learning and  proposes a novel method, LPI, which extends the prompt learning for multi-modal vision-langugae retrieval in the continual learning setting.
Overall, the paper is well written and easy to follow. Extensive experiments are conducted to show the effectiveness of the proposed method.

**Limitations:**

1. The second paragraph of the introduction, the references should be cited when the authors talk about the related work.
2. The resolution of Fig.2 needs improvement. And for Task i-1, it's better to add lines between the model and CPA module, just like the Task i.
3. In Line 412, it missed the braces.
4. For the cross-task interaction, how do you choose the predefined threshold of the task?
And what is the meaning of "Pairs of tasks with cosine similarity greater than and equal to a predefined threshold are considered positive tasks, while those below the threshold are treated as negative tasks,".
From my understanding, during training, the task identity is already known, why the threshold is still needed?
5. For the ablation study in Table 3, it's interesting that why CP is much lower than DP. CP without compression should contain more information.

**Suitability:**

3

---

### Official Review · Reviewer_hLWf · 2024-05-24

**Rating:** 4
**Confidence:** 2

**Summary:**

This paper introduces a LPI method for continual vision-language retrieval tasks. The proposed LPI enables effective cross-modal and cross-task interaction with a low-rank interaction decomposition module to address the parameter explosion issue in multimodal understanding. Additionally, the hierarchical low-rank contrastive learning strategy enhances training robustness, ensuring prompt alignment across different layers.

**Strengths:**

1.The proposed LPI effectively addresses the key issue in current multimodal understanding by combining cross-modal and cross-task interactions.
2.The low-rank interaction decomposition module overcomes the parameter explosion problem caused by the increasing number of tasks.
3. The extensive experiments on image-text retrieval and referring expression comprehension demonstrate the efficacy of LPI.

**Limitations:**

1. The paper lacks a detailed analysis of computational complexity and scalability. While the method optimizes parameter quantity, it does not discuss computational complexity in depth, which may hinder its implementation in larger and more complex tasks.
2. The necessary comparative methods in Table.1 are lacking:
[1] Multilateral Semantic Relations Modeling for Image Text Retrieval, CVPR '23
[2] HGAN: Hierarchical Graph Alignment Network for Image-Text Retrieval, TMM 2023
[3] MKVSE: Multimodal Knowledge Enhanced Visual-semantic Embedding for Image-text Retrieval, TOMM 2023
[4] External Knowledge Dynamic Modeling for Image-text Retrieval, ACMMM '23
3. The source code is not available.

**Suitability:**

3

---

### Official Review · Reviewer_FYp2 · 2024-05-24

**Rating:** 4
**Confidence:** 2

**Summary:**

The paper introduces Low-rank Prompt Interaction (LPI) for multi-modal understanding, integrating cross-modal and cross-task interactions. It incorporates multi-modal correlation modules in Transformer layers and utilizes Low-rank Interaction-augmented Decomposition to manage memory and enhance cross-modal ties without adding excessive parameters. Hierarchical low-rank contrastive learning improves robustness, while task-specific contrasts are used to tailor learning processes based on task differences. The method demonstrates improved performance on two retrieval tasks, confirming its effectiveness with minimal parameter increase.

**Strengths:**

1. The paper is well-written and easy to follow.
2. The visualizations are pretty.
3. Good experimental results on several scenarios.

**Limitations:**

1. It is suggested that future revisions include a deeper exploration of the motivations behind specific design decisions, enhancing the reader's understanding of the reasons for these choices beyond just a description of the implemented actions.
2. I see the overall loss is a combination of several losses, how do authors balance the contribution of each loss?

**Suitability:**

2

---

### Meta-Review · Area_Chair_gH6U · 2024-06-29

**Recommendation:** Accept (Poster)
**Confidence:** 4

**Metareview:**

The paper introduces Low-rank Prompt Interaction (LPI) for multi-modal understanding, which integrates cross-modal and cross-task interactions via multi-modal correlation modules in Transformer layers. It employs Low-rank Interaction-augmented Decomposition to balance memory efficiency and cross-modal connectivity with minimal parameter increase. Enhanced by hierarchical low-rank contrastive learning and task-specific contrasts, LPI shows improved performance on two retrieval tasks, demonstrating its effectiveness.